# Role of Delocalization, Asymmetric Distribution of π-Electrons and Elongated Conjugation System for Enhancement of NLO Response of Open Form of Spiropyran-Based Thermochromes

**DOI:** 10.3390/molecules28176283

**Published:** 2023-08-28

**Authors:** Naveen Kosar, Saba Kanwal, Malai Haniti S. A. Hamid, Khurshid Ayub, Mazhar Amjad Gilani, Muhammad Imran, Muhammad Arshad, Mohammed A. Alkhalifah, Nadeem S. Sheikh, Tariq Mahmood

**Affiliations:** 1Department of Chemistry, University of Management and Technology (UMT), C-11, Johar Town, Lahore 54770, Pakistan; 2Department of Chemistry, COMSATS University Islamabad, Abbottabad Campus, Abbottabad 22060, Pakistan; 3Chemical Sciences, Faculty of Science, Universiti Brunei Darussalam, Jalan Tungku Link, Gadong BE1410, Brunei; 4Department of Chemistry, COMSATS University Islamabad, Lahore Campus, Lahore 45550, Pakistan; 5Department of Chemistry, Faculty of Science, King Khalid University, Abha 61413, Saudi Arabia; 6Institute of Chemistry, The Islamia University of Bahawalpur, Baghdad-ul-Jadeed Campus, Bahawalpur 63100, Pakistan; 7Department of Chemistry, College of Science, King Faisal University, Al-Ahsa 31982, Saudi Arabia; 8Department of Chemistry, College of Science, University of Bahrain, Sakhir P.O. Box 32038, Bahrain

**Keywords:** thermochromes, spiropyran, nonlinear optics, refractive index, DFT

## Abstract

Switchable nonlinear optical (NLO) materials have widespread applications in electronics and optoelectronics. Thermo-switches generate many times higher NLO responses as compared to photo-switches. Herein, we have investigated the geometric, electronic, and nonlinear optical properties of spiropyranes thermochromes via DFT methods. The stabilities of close and open isomers of selected spiropyranes are investigated through relative energies. Electronic properties are studied through frontier molecular orbitals (FMOs) analysis. The lower HOMO-LUMO energy gap and lower excitation energy are observed for open isomers of spiropyranes, which imparts the large first hyperpolarizability value. The delocalization of π-electrons, asymmetric distribution and elongated conjugation system are dominant factors for high hyperpolarizability values of open isomers. For deep understanding, we also analyzed the frequency-dependent hyperpolarizability and refractive index of considered thermochromes. The NLO response increased significantly with increasing frequency. Among all those compounds, the highest refractive index value is observed for the open isomer of the spiropyran **1** (1.99 × 10^−17^ cm^2^/W). Molecular absorption analysis confirmed the electronic excitation in the open isomers compared to closed isomers. The results show that reversible thermochromic compounds act as excellent NLO molecular switches and can be used to design advanced electronics.

## 1. Introduction

Since the 1970s, the designing and synthesis of thermochromic materials have been an important interdisciplinary research field because of their excellent efficiency in changing their characteristics concerning temperature [1,2]. Several applications of thermochromic materials have been reported, such as in cosmetics [3], as food quality indicators, forehead thermometers [4,5], thermal mapping, refrigeration thermometers, thermochromic sensors [6,7], memory devices and cholesteric liquid crystals (which are used in biomedical therapy) [8] etc.

Thermochromes are classified as reversible or irreversible [9]. Irreversible thermochromic materials cannot return to their original state [10]. Although there are several applications of irreversible thermochromic materials in different fields but in most cases, thermochromism reversibility is considered a vital requirement. Crystal violet lactone is the most essential class of reversible thermochromism [11]. Towns et al. reported reversible spiropyranes. Those convert from colorful to colorless complexes with respect to changes in temperature [12]. Spiropyranes are extensively studied due to their structural variability and clear color change under temperature variations [13], upon heating, the closed isomer of spiropyran changes to an unstable open isomer having a specific color. The color change depicts the electronic excitation in these thermochromes due to extended conjugation and unequal electronic distribution.

The literature revealed the generation of optical and NLO properties of compounds if electronic excitations occur [14,15,16,17,18,19]. For example, the NLO response of photochemical/thermochemical interconversion of substituted dihydroazulene (DHA) to vinylheptafuluene (VHF) was analyzed by Pozzo and coworkers. Electron withdrawing group (NO_2_) has a pronounced effect on the first hyperpolarizability compared to the electron donating group (NH_2_) [20]. Previtali et al. synthesized different polymorphs of cyclic triimidazle pyrene bio probe. They observed that they have tunable linear and nonlinear optical properties where one of the polymorphs TTPyr(HT), obtained at high temperature, has ten times higher second harmonic generation efficiency than urea [21]. Sliwa et al. synthesized *N*-(4-hydroxy)-salicylidene-amino-4-(methylbenzoate) and *N*-(3,5-di-tert-butylsalicylidene)-4-aminopyridine. Upon irradiation with UV light using TD-DFT analysis, they change color and possess better NLO response than standard urea [22]. Photochromic interconversion between spiropyran and merocyanine results in a large light switchable change in the second and third-order nonlinearity [23,24].

Le-Bozec and co-workers thoroughly studied the photochromic transformation of 4,4′-bis(ethenyl)-2,2′-bipyridine ligands functionalized by a dimethylaminophenyl–dithienylethene group and the corresponding zinc (II) complex. The compound efficiently undergoes reversible interconversion into open and closed forms, where the closed form shows outstanding NLO response due to π-conjugation (which is absent in open form) [25]. They also designed several switching photochromes with excellent ON/OFF photoswitchable NLO response [26]. Jing et al. computationally explored the NLO response of photochromic dihydroazulene/vinylheptafulvene with Ru-based metal complexes. They observed the largest second-order nonlinear response for an open form of these complexes which was further increased with frequency [27]. This detailed survey illustrates several synthetic and computational reports on reversible NLO photoswitches in literature [28]. But computational work on reversible NLO thermoswitches is not explored yet. Keeping in view the importance of thermochromic materials, in this study, we have performed a DFT analysis of the NLO response of open and closed isomers of Spiropyranes-based reversible thermochromes (See Figure 1).

## 2. Computational Details

All calculations were performed by using Gaussian-09 software [31]. The selected structures of compounds were optimized at ωB97XD/6-311+G(d, p) method. The selection of ωB97XD density functional captures long-range and short-range interactions in a system and intersystem charge transfer property. The Pople’s 6-311+G (d, p) basis set gives an idea about the localization of electronic density in orbitals [32,33,34]. After optimization, frequency calculations are performed at the same level of theory to verify true optimization, which is confirmed by the absence of any imaginary frequency. Thus, after obtaining the optimized geometries of thermochromic compounds, we investigated the relative stability of open and closed forms by considering the energy difference (closed/open isomer) at the same level of theory. Moreover, for all other properties, ωB97XD/6-311+G (d, p) method is used. GaussView 5.0 is used for the visualization of structures [35]. 

NBO charge analysis is also performed to know the charge transfer in molecules [36]. Frontier molecular orbital (FMO) analysis was performed to calculate molecular species’ electronic stability and chemical reactivity. The dipole moment of open and closed isomers was calculated at above mention level of theory. TD-DFT calculations were carried out at ωB97XD/6-311+G(d, p) method to determine the spectrum region where these thermochromic compounds show absorbance and transparency. Moreover, these thermochromic compounds’ nonlinear optical (NLO) properties of open and closed isomers were calculated at the ωB97XD/6-311+G(d, p) method. ωB97XD is a long-range corrected functional method and is best for calculating NLO properties [37,38,39,40]. Linear optical property polarizability (*α*_o_) and nonlinear optical properties (static first hyperpolarizability (*β*_o_), frequency-dependent first, second hyperpolarizability and reflective index were calculated at the above-mentioned level of theory.

Mathematical expressions for polarizability (*α*_o_) and static first hyperpolarizability (*β*_o_) are given below: *α*_o_ = 1/3(*α*_xx_ + *α*_yy_ + *α*_zz_)(1)
*β*_o_ = [*β*_x_^2^ + *β*_y_^2^ + *β*_z_^2^]^½^(2)
where,
*β*_x_ = *β*_xxx_ + *β*_xxy_ + *β*_xxz_, *β*_y_ = *β*_yyy_ + *β*_yzz_ + *β*_yxx_ and *β*_z_ = *β*_zzz_ + *β*_zxx_ + *β*_zyy_

The frequency-dependent nonlinear optical analysis was investigated at 5.63 × 10^14^ Hz (0.09 au) and 2.82 × 10^14^ Hz (0.04 au). Frequency-dependent hyperpolarizability contains the approximation of electro-optic Pockel’s effect *β*(−*ω*;*ω*,0) and second harmonic generation *β*(−2*ω*; *ω*, *ω*). For estimation of frequency-dependent second hyperpolarizability (*γ*), dc-Kerr effect (*ω*) = γ(−*ω*; *ω*,0,0) and electric field induced second harmonic generation (*ω*) = γ(−2*ω*; *ω*, *ω*,0) and degenerate four-wave mixing are calculated. These values were also used to estimate the refractive index of the respective compounds.

## 3. Results and Discussion 

### 3.1. Geometric and Structural Properties of Close and Open Isomers of Spiropyranes

The geometric stability of close and open isomers of spiropyranes is estimated from their geometric properties. The optimized geometries are given in Figure 2, and the geometric parameters of spiropyranes are given in Table 1. Optimized geometries of spiropyran **1** show a clear change in bond distance between the ring closing atoms (C-O). In the close isomer of spiropyran **1**, the ring-closing carbon and oxygen atoms (C-O) bond length is 1.43 Å. This bond distance increases as the close structure is converted to an open form. The C-O bond length of ring closing atoms is increased to 4.11 Å, which confirms the breakage of the C-O bond. Similar behavior is observed for spiropyran **2**, where the bond length between ring closing C and O atoms is increased from closed isomer to open isomer. The optimized structure of the close isomer of spiropyran **2** shows terminal C-O atoms bond length of 1.36 Å. In contrast, this terminal C-O atoms bond length in an open isomer increases to 4.30 Å. These results are comparable to the X-rays data of the C-O bond length of spyropyran reported by Ozhogin et al. [29].

The relative energies are calculated to check the thermodynamic stability for close and open isomers of spiropyranes (**1** & **2**) and are given in Table 1. Relative energy measures the energy difference between the most stable isomer and another. In spiropyran **1**, the close isomer is thermodynamically more stable than the open isomer of the spiropyran. The open isomer is 5.78 kcal mol^−1^ less stable than the closed isomer. Menzonatto and Lopes studied the mechanistic pathway of spiropyrane close-form conversion to open-form, and they observed more stability for close compared to open-form [41]. In spiropyran **2**, the open isomer is thermodynamically more stable (by 0.29 kcal mol^−1^) than the close isomer of spiropyran **2** (see Table 1). We also calculated the activation barrier for the thermal opening of both spiropyranes **1** & **2**. The activation barriers for spiropyranes **1** & **2** are 12.74 kcal mol^−1^and 14.83 kcal mol^−1^, respectively. The structures of transition states are given in Appendix A. The reported activation barrier for C_spiro_-O bond cleavage is 7.6 kcal mol^−1^. The high activation barrier of spiropyranes is due to their rotation from close to open isomer during bond cleavage. The rotation is necessary to elongate the C_spiro_-O bond and a larger torsional angle after cleavage of this bond [42].

### 3.2. NBO Charge Analysis of Close and Open Isomers of Spiropyranes (**1** & **2**)

The NBO charges of close and open isomers of spiropyranes (**1** & **2**) are carried out to estimate the acceptor and donor sites of the closing terminal carbon and oxygen (C-O) atoms, and the results are given in Table 1. In the open isomer of the spiropyran **1,** the NBO charge value obtained for the carbon atom is 0.64 |e| and for the oxygen atom is −0.63 |e|. In spiropyran **2**, the charge on terminal carbon is −0.05 |e| and −0.65 |e| is observed on the oxygen atom in open form. The opposite charges on terminal carbon and oxygen atoms in open isomers of spiropyranes **1** & **2** depict the possibility of electrostatic interactions. In a close isomer of **1**, the charge on the ring closing oxygen atom is −0.59 |e| and on the carbon atom is 0.75 |e|, respectively. In the close isomer of spiropyran **2**, ring-closing oxygen and carbon atoms have −0.57 |e| and 0.37 |e| charges, respectively. Variations in charges of terminal (ring closing) oxygen and carbon atoms indicate the electrostatic interaction that results in bond formation between terminal carbon and oxygen atoms in individual spiropyranes (**1** & **2**). These opposite charges on terminal carbon and oxygen atoms in close isomers represent the polarity of the bonds. Loopes and coworkers also theoretically designed spiropyranes derivatives, and their NBO data is similar to our results [41]. 

### 3.3. Dipole Moment of Close and Open and Isomers of Spiropyranes (**1** & **2**)

The separation of charges in a system is defined as the dipole moment. More separation of charges results in a higher dipole moment. The NBO results indicate a similar trend of increasing dipole moment for both spiropyran isomers (**1** & **2**). The open forms of both spiropyranes (**1** & **2**) have a higher dipole moment than the close form. In spiropyran **1,** the close isomer has a lower dipole moment of 6.49 D than the open isomer (8.78 D). In the case of spiropyran **2**, the open isomer has a higher dipole moment of 10.23 D than 6.26 D of close form. The variation in dipole moments is due to the larger internuclear distance between oppositely charged carbon and oxygen, along with large quantities of charges on these terminal atoms (C and O), as discussed vide supra. The difference in charges of spiro-carbon and oxygen is more for open isomers than closed isomers, and distances are also more for open isomers. As a result (open isomers of spiropyranes **1** & **2**) dipole moment is also higher. Dipole moment values for spiropyranes are reported in Table 2. The higher dipole moment of the open isomers of spiropyranes (**1** & **2**) illustrates that these could be good candidates for generating high NLO response.

### 3.4. Frontier Molecular Orbitals (FMOs) Analysis of Close and Open Isomers of Spiropyranes (**1** & **2**)

After calculating the geometric parameters, FMOs analysis is executed to observe the electronic stability and reactivity of open and closed isomers of spiropyranes (**1** & **2**). The energies of HOMOs (E_H_), the energies of LUMOs (E_L_) and the HOMO-LUMO gap (E_H-L_) of spiropyranes (**1** & **2**) are given in Table 2. E_H-L_ of a close isomer of spiropyran **1** is 8.36 eV which is noticeably higher than the open isomer (6.34 eV). Similarly, the close isomer of spiropyran **2** has an E_H-L_ of 7.51 eV, more than the open isomer of spiropyran **2** (4.26 eV). The reduction of E_H-L_ in open isomers spiropyras (**1** & **2**) illustrates the shifting of electronic density in these conjugated systems, which is related to the conductivity. The lower E_H-L_ of a system has more charge transfer, generating conductive properties. The close isomers of both spiropyranes **1** & **2** indicate insulator properties due to large E_H-L_ converted to semi-conductors in open isomer. The isodensities of HOMOs, LUMOs and their energy gaps (E_H-L_) of close and open isomers of spiropyranes are shown in Figure 3. A similar kind of variation in E_H-L_ values was observed experimentally by Kudernac et al. during their work on nano-electronic switches [43].

The HOMOs densities are distributed on the whole skeleton except the methoxy group of close isomers of spiropyran **1**. The LUMO densities are localized on the central benzopyran ring of a close isomer of spiropyran **1**. The HOMOs densities are distributed on the skeleton of an open isomer of spiropyran **1** but are more concentrated on the phenolic ring. The LUMO densities are delocalized on the entire skeleton in this isomer. For close isomers of spiropyran **2,** the HOMOs densities are distributed on the aromatic amine. The LUMOs densities are localized on benzofuran. The HOMOs densities of an open isomer of spiropyran **2** are distributed on the whole skeleton, but more densities are present on the aromatic amine. The LUMOs densities of this open isomer are localized on acetic acid-containing rings in this isomer. Comparatively, between open and closed isomers, the HOMOs densities are more delocalized in the case of open isomers of both spiropyranes **1** & **2**. 

### 3.5. Linear and Nonlinear Optical (NLO) Properties of Close and Open Isomers of Spiropyranes (**1** & **2**)

#### 3.5.1. Polarizability (α_o_) and First Hyperpolarizability (β_o_) Analyses

Linear and nonlinear optical parameters are calculated after geometric and electronic analyses. Polarizability (*α*_o_) and the first hyperpolarizability (*β*_o_) of close and open isomers of spiropyranes **1** & **2** are given in Table 2, and a graphical representation is given in Figure 4. 

The polarizability value of close and open isomers of spiropyran **1** is 276 au and 350 au, respectively. Similarly, for close and open isomers of spiropyran **2**, *α*_o_ is 413 au and 460 au, respectively. Open isomers of the spiropyranes show comparatively high polarizability than close isomers. The first hyperpolarizability (*β*_o_) results of both isomers show a prominent increase in hyperpolarizability value from close to open isomers of both **1** and **2**. The *β*_o_ of close and open isomers of the spiropyrane **1** is 231.89 au and 2381.11 au, respectively. Bond length alternation and asymmetric electronic density distribution affect *β*_o_ of an open isomer of spiropyran **1** absent in a closed isomer. The *β*_o_ value for the close isomer of spiropyran **2** is 1424.51 au and the open isomer has *β*_o_ value is 3594.49 au. The hyperpolarizability response of the open isomer of spiropyran **2** is larger due to extended conjugation (delocalization of electrons), which is absent in the close isomer. 

The NLO response of close and open isomers of spiropyranes **1** & **2** are also observed in acetonitrile solvent, and their results are given in Table 2 To see the difference in the gas phase and aqueous phase effect on open and closed isomers of spiropyranes **1** & **2**, we performed NLO calculations in this solvent. As the previous reports illustrate that reversible spiropyranes show high thermal stability in acetonitrile we select this solvent in our study [44]. The *β*_o_ of close and open isomers of the spiropyran **1** are 1242.20 au and 4876.61 au, respectively. The *β*_o_ values increase in the solvent phase.

The *β*_o_ of close and open isomers of the spiropyran **2** are 2844.07 au and 3578.72 au, respectively. The *β*_o_ values increase in the solvent phase for the close isomer, but for the open isomer, it is a little bit decreased. Similar to spiropyrane **1**, the *β*_o_ values of open isomers are higher in comparison to close isomers. The increasing trend of *β*_o_ values from close to open form of both spiropyranes **1** & **2** is similar to gas phase calculations. Following is an increasing trend of *β*_o_ values; open isomer-spiropyran **1** (4876.61 au) > open isomer-spiropyran **2** (3578.72 au) > close isomer-spiropyran **2** (2844.07 au) > close isomer-spiropyran **1** (1242.20 au). The increase in hyperpolarizability is increased in the dipole moment and polarizability of these isomers, as given in Appendix A.

#### 3.5.2. Two-Level Model Analysis of Close and Open Isomers of Spiropyranes **1** & **2**

Two level model is implemented to investigate the internal parameters responsible for the increase in the first hyperpolarizability value of closed and open isomers of spiropyranes **1** & **2**. These parameters include excitation energy (ΔE), oscillating strength (ƒ_o_) and excited dipole moment (Δ*µ*), and their results are summarized in Table 3. The excitation energy of close isomers of spiropyranes **1** & **2** is 6.40 eV and 4.62 eV, respectively. The ΔE value is lower for open isomers of both spiropyranes **1** (2.97 eV) & **2** (4.55 eV). The trend of decreasing ΔE values from close to open isomers is inconsistent with an increase in *β*_o_ values. The *β*_o_ values of open isomers **1** & **2** are higher, and their excitation energies are also lower. Moreover, the ƒ_o_ and Δ*µ* values also show a similar increasing trend compared to *β*_o_ values moving from close to open isomers of spiropyran **1** & **2**. The ƒ_o_ (Δ*µ)* values of open isomers of spiropyranes **1** and **2** are 0.68 (8.78 Debye) and 0.53 (10.22 Debye), respectively. The ƒ_o_ (Δ*µ)* values of close isomers of spiropyranes **1** and **2** are 0.54 (6.49 Debye) and 0.41 (6.26 Debye), respectively. In addition, it is confirmed that the dipole moment values do not show any change between the ground and the excited states in the gas phase. The variational trend of *β*_TLM_ values is comparable to the trend of increasing *β*_o_ values. So, ΔE, ƒ_o_ and Δ*µ* are the decisive factors for the enhancement of NLO response in these isomers.

The internal parameters are also observed in the presence of acetonitrile solvent for closed and open isomers of spiropyranes **1** & **2**, and their results are described in the Section 3.6 (Table 6, *vide infra*). The ΔE and Δ*µ* of spiropyranes **1** & **2** in the solvent phase are similar to gas phase results. The ΔE value (5.03 eV) of the open isomer is lower, and Δ*µ* value (11.74) is higher. The *β*_o_ value (2381.11 au) of this isomer is also higher. The ΔE value (5.26 eV) of the close isomer is lower, and Δ*µ* value (11.74 Debye) is higher and *β*_o_ value of this isomer is also low (231.89 au). The trend of increasing ƒ_o_ is opposite to the *β*_o_ value for closed and open isomers. Similar results are obtained for spiropyran **2** as obtained for spiropyran **1**. The ΔE value (4.60 eV) of the open isomer is lower, and ƒ_o_ (0.89) & Δ*µ* value (13.91 Debye) are higher, which is according to the two-level model. Alternatively, the ΔE value (4.57 eV) of the open isomer is lower, and ƒ_o_ (0.62) & Δ*µ* value (8.36 Debye) are higher. Overall, excitation energy and variational dipole moment are decisive factors in describing the NLO response of these systems.

#### 3.5.3. Frequency-Dependent First Hyperpolarizability of Close and Open Isomers of Spiropyranes **1** & **2**

Frequency-dependent first hyperpolarizability is the fundamental parameter in the NLO phenomenon. Experimentalists required frequency-dependent first hyperpolarizability information for the development of optical devices. These properties give guidelines for the behavior of NLO materials when they are treated with light beams of different frequencies in practice. The second harmonic generation is one of the effects governed by hyperpolarizabilities [45]. Here, the frequency-dependent first hyperpolarizability is calculated for spiropyranes **1** & **2,** and results are given in Table 4. All the calculations are implemented at routinely used wavelengths of 5.63 × 10^14^ Hz (0.09 au) and 2.82 × 10^14^ Hz (0.04 au) [46]. The electro pockel effect *β*(−ω;ω,0) values for close and open isomers of spiropyran **1** are in the range of 255.9 to 6324.6 au at a wavelength of 5.63 × 10^14^ Hz (0.09 au) and 2.82 × 10^14^ Hz (0.04 au), respectively. The close and open isomers of spiropyran **1** showed the most prominent second harmonic generation effect *β*(−2ω;ω,ω) values in the range of 317.9–6698.7 au at 5.63 × 10^14^ Hz (0.09 au) and 2.82 × 10^14^ Hz (0.04 au), respectively. The higher electro-pockel effect value (6324.6 au) and second harmonic generation effect value (6698.7 au) are obtained for the open isomer of spiropyran **1** at 5.63 × 10^14^ Hz (0.09 au). These results are similar to the work of Tonnele and Castet [47].

The electro pockel effect *β*(−ω;ω,0) values for close and open isomers of spiropyran **2** range from 1565.4 au to 22,805.2 au at a wavelength of 5.63 × 10^14^ Hz (0.09 au) and 2.82 × 10^14^ Hz (0.04 au). Again, the close and open isomers of spiropyran **2** showed prominent second harmonic generation effect *β*(−2ω;ω,ω) values from 1893.9–54,124.8 au at 5.63 × 10^14^ Hz (0.09 au) and 2.82 × 10^14^ Hz (0.04 au), respectively. The higher electro-optical pockel effect value (22,805.2 au) and second harmonic generation effect value (54,124.8 au) are obtained for the open isomer of spiropyran **2** at 5.63 × 10^14^ Hz (0.09 au). The electro-optical Pockel effect and second harmonic generation are highly affected at the wavelength of 5.63 × 10^14^ Hz (0.09 au) for both open and closed isomers of spiropyran **2**. Among spiropyranes **1** & **2**, the highest electro-optical pockel effect and second harmonic generation effect values are seen for the open isomer of spiropyran **2**. These results are comparable to frequency-dependent work on other photoswitches [47].

#### 3.5.4. Frequency-Dependent Second Hyperpolarizability and Quadratic Nonlinear Refractive Index Analysis of Spiropyranes **1** & **2**

The second-order electric susceptibility per unit volume is known as the second hyperpolarizability. Experimentalists required frequency-dependent second hyperpolarizability for practical applications where the hyperpolarizability is calculated at different wavelengths [48]. The frequency-dependent second hyperpolarizability results for close and open isomers of spiropyranes **1** & **2** are given in Table 5. All the calculations are implemented at routinely used wavelengths of 5.63 × 10^14^ Hz (0.09 au) and 2.82 × 10^14^ Hz (0.04 au) [46]. It is observed that the gamma (*γ*) values for close and open isomers of spiropyran **1** at static frequency are 3564.7 au and 6907.9 au, respectively. The large second hyperpolarizability (6907.9 au) value is obtained for the open isomer of the spiropyran **1**. The dc-Kerr effect values for close isomer of the spiropyran **1** are 5095.1 au and 3909.5 au at 5.63 × 10^14^ Hz (0.09 au) and 2.82 × 10^14^ Hz (0.04 au), respectively. The EFSHG values for close isomers are 3037.8 au and 4636.6 au at frequencies of 5.63 × 10^14^ Hz (0.09 au) and 2.82 × 10^14^ Hz (0.04 au), respectively. The dc-Kerr effect values for an open isomer of the spiropyran **1** are 52,842.8 au and 9030.3 au at a wavelength of 5.63 × 10^14^ Hz (0.09 au) and 2.82 × 10^14^ Hz (0.04 au), respectively. The open isomer of spiropyran **1** also shows the prominent electric field-induced second harmonic generation effect values of 2398.2 au and 23,299.4 au at 5.63 × 10^14^ Hz (0.09 au) and 2.82 × 10^14^ Hz (0.04 au), respectively. The high values of the dc-Kerr effect coefficient are observed for open isomers of the spiropyran **1** (52,842.8 au) & **2** (126,583.9 au) at 5.63 × 10^14^ Hz (0.09 au). The higher values of EFSHG coefficient are seen for open isomers of the spiropyran **1** (23,299.4 au) & **2** (33,722.1au) at 2.82 × 10^14^ Hz. The highest frequency-dependent dc-Kerr effect coefficient value (126,583.9 au) is obtained for open isomers of the spiropyran **1** at 5.63 × 10^14^ Hz (0.09 au). These results are similar to the reported data for acid-controlled second-order nonlinear optical switches [49]. We observed that the delocalization of π electrons, elongated conjugation system and asymmetric distribution of electrons are dominant factors in increasing static and frequency-dependent first and second hyperpolarizability response of open isomers of spiropyranes **1** & **2**.

Static second hyperpolarizability (*γ*(0; 0, 0, 0)) values for close and open isomers of spiropyran **2** are 11,533.7 au and 27,098.8 au, respectively. The large *γ*_o_ value is obtained for the open isomer of the spiropyran **2**. The dc-Kerr effect values for close isomer of spiropyran **2** are 20,011.3 au and 13,033.1 au at 5.63 × 10^14^ Hz (0.09 au) and 2.82 × 10^14^ Hz (0.04 au), respectively. The EFSHG values for close isomers are 1027.6 au and 1713.9 au. At 5.63 × 10^14^ Hz (0.09 au) and 2.82 × 10^14^ Hz (0.04 au), the dc-Kerr effect values for an open isomer of spiropyran **2** are 126,583.9 au and 120,760.7 au, respectively. The open isomer of spiropyran **2** also shows the noticeable electric field-induced second harmonic generation effect values, which are 112,874.0 au and 33,722.1 au at 5.63 × 10^14^ Hz (0.09 au) and 2.82 × 10^14^ Hz (0.04 au), respectively. The higher dc-Kerr effect value (126,583.9 au) is detected for both isomers of spiropyran **2** at 5.63 × 10^14^ Hz (0.09 au). The high EFSHG effect value (33,722.1 au) is detected for both isomers of spiropyran **2** at 2.82 × 10^14^ Hz.

The refractive index (*n*_2_) values obtained for a close isomer of the spiropyran **1** are 3.23 × 10^−18^ cm^2^/W and 1.01 × 10^−18^ cm^2^/W at 5.63 × 10^14^ Hz (0.09 au) and 2.82 × 10^14^ Hz (0.04 au), respectively. The open isomer of the spiropyran **1** has refractive index values of 1.99 × 10^−17^ cm^2^/W and 3.25 × 10^−18^ cm^2^/W at a wavelength of 5.63 × 10^14^ Hz (0.09 au) and 2.82 × 10^14^ Hz (0.04 au), respectively. Among both isomers (close and open) of spiropyran **1**, the open isomer has the large *n*_2_ (1.99 × 10^−17^ cm^2^/W) value at the wavelength of 5.63 × 10^14^ Hz (0.09 au). The refractive index values found for close isomer of spiropyran **2** are 1.11 × 10^−17^ cm^2^/W and 3.45 × 10^−18^ cm^2^/W. At 5.63 × 10^14^ Hz (0.09 au) and 2.82 × 10^14^ Hz (0.04 au), the open isomer of spiropyran **2** has refractive index values of 2.21 × 10^−17^ cm^2^/W and 1.50 × 10^−17^ cm^2^/W at 5.63 × 10^14^ Hz (0.09 au) and 2.82 × 10^14^ Hz (0.04 au), respectively. The highest refractive index value (2.21 × 10^−17^ cm^2^/W) is observed at the wavelength of 5.63 × 10^14^ Hz (0.09 au) for open isomer of spiropyran **2** at 5.63 × 10^14^ Hz (0.09 au).

### 3.6. Absorption Spectra for Close and Open Isomers of Spiropyranes **1** & **2**

The absorption spectra of both isomers of spiropyranes **1** & **2** have significant absorption in the entire UV region and somewhat in the visible region of the electromagnetic spectrum. The λ_max_ values are summarized in Table 3 and graphically represented in Figure 5. It is found that the close isomer of the spiropyran **1** showed absorbance in the UV region at 194 nm, while the open isomer of the spiropyran **1** showed maximum absorbance in the visible region at 418 nm. For spiropyran **2**, both closed, and open isomers of the compound show maximum absorbance in the UV region at 269 nm and 272 nm, respectively. Both isomers (close and open) of spiropyran **1** & **2** show transparency in the deep UV region (below 200 nm), except for the close isomer of spiropyran **1**. The increase in λ_max_ values for open isomers supports the NLO response. As the electronic excitation increases in a system, the NLO response also increases. Such behavior is previously seen by Bibi et al. for on-off switches based on cobalt and iron-containing coordinate complexes [50], which justifies our results. 

The UV-Vis analysis of close and open isomers of spiropyranes **1** & **2** are also performed in acetonitrile solvent, and their results of λ_max_ are given in Table 6. The λ_max_ for the open isomer (246 nm) is higher than the closed isomer (236 nm) of spiropyran **1**. In continuation to this result, the λ_max_ of _the_ close isomer (271 nm) is higher than the open isomer (270 nm) of spiropyran **2**. Although compared to gas phase calculations, the λ_max_ of closed isomer is increased, and for open isomer, this value is decreased significantly. The λ_max_ values in acetonitrile are comparable to already reported λ_max_ values reported by spiropyranes in the same solvent [51].

### 3.7. The Charge-Transfer Distance or Density Difference between Ground and Excited States for Close and Open Isomers of Spiropyranes **1** & **2**

The charge transfer distance and charge density difference of both spiropyranes **1** & **2** isomers are calculated with Multiwfn software [52]. Figure 6 shows the charge density difference of the excited states where the green and blue densities correspond to the region where the electron density increases and decreases after electron excitation, respectively. The change in the position of electronic densities, HOMO (electron) and LUMO (hole) in Figure 6 compared to Figure 3 for the ground state density shoe the shifting of charge from one place to another. The integral of overlap of hole-electron (S) and the distance between the centroid of the hole and electron (D in Å) for close and open isomers of spiropyranes **1** & **2** to five excited states are given in Table 7. According to the rule, the excited states with a high D value and lower S value have Charge transfer mode (CT type) and those excited states with lower D values and large S values are Local excitation type (LE type). When both D and S values are small, they are Rydberg-type excitation. The close isomer of spiropyran **1**, excited states 1 (D/S = 1.57/0.21) and 5 (D/S = 2.07/0.17), have high D-values and lower S values, showing CT-type excitations. The rest of the excited states 2–4 must have LE-type excitations. The open isomer of spiropyran **1**, excited states 2 (D/S = 2.36/0.15) and 5 (D/S = 2.84/0.27) have high D-values and lower S values, showing CT-type excitations. The rest of the excited states 1,3, and 4 have LE-type excitations. The close isomer of spiropyran **2**, excited states 1 (D/S = 1.94/0.26) and 4 (D/S = 2.51/0.20), have high D and lower S values, showing CT-type excitations. The rest of the excited states 2, 3, and 5 have LE-type excitations. The open isomer of spiropyran **2**, excited states 3 (D/S = 2.43/0.25) and 4 (D/S = 2.98/0.34), have high values of D values and lower S values, so they show CT-type excitations. The rest of the excited states, 1 and 5, have LE-type excitations. Compared to other states, the excited state 2 shows Rydberg-type excitation as both D (1.73 Å) and S (0.21) values are small.

## 4. Conclusions

We have investigated the geometric, electronic, optical, and nonlinear optical properties of reversible thermochromic spiropyranes at the DFT method. Electronic properties of these close and open isomers of spiropyranes **1** & **2** are studied through FMOs analysis. It is observed that the lower HOMO-LUMO gap with lower excitation energy in open isomers of both spiropyranes **1** & **2** imparts large hyperpolarizability values. The extended electronic delocalization and unsymmetrical electronic distribution are responsible for the decrease in the HOMO-LUMO gap. Comparatively, the first hyperpolarizability values of open isomers are more than the closed isomers. The delocalization of π-electrons, asymmetric distribution of electrons (in spiropyran **1**) and elongated conjugation system (in spiropyran **2**) are dominant factors for the increase of hyperpolarizability values of open isomers (**1** & **2**). In both spiropyranes, the highest *β*_o_ (3594.49 au) is observed for the open isomer of spiropyran **2**. To get a deep understanding of our studied reversible thermochromic spiropyranes (**1** & **2**), we also analyzed the frequency-dependent hyperpolarizability. The more prominent frequency-dependent NLO response is at 5.63 × 10^14^ Hz (0.09 au). To further observe their NLO response, we also calculated the refractive index of these thermochromic compounds. Among all those compounds, the highest refractive index value is observed for the open isomer of the spiropyran **1** (1.99 × 10^−17^ cm^2^/W). Molecular absorption analysis confirmed electronic excitation in open and closed isomers, where the effect is more pronounced for open isomers of thermochromic spiropyranes (**1** & **2**). Therefore, the calculated values reveal reversible thermochromic compounds as excellent NLO molecular switches for future applications.

## Figures and Tables

**Figure 1 molecules-28-06283-f001:**
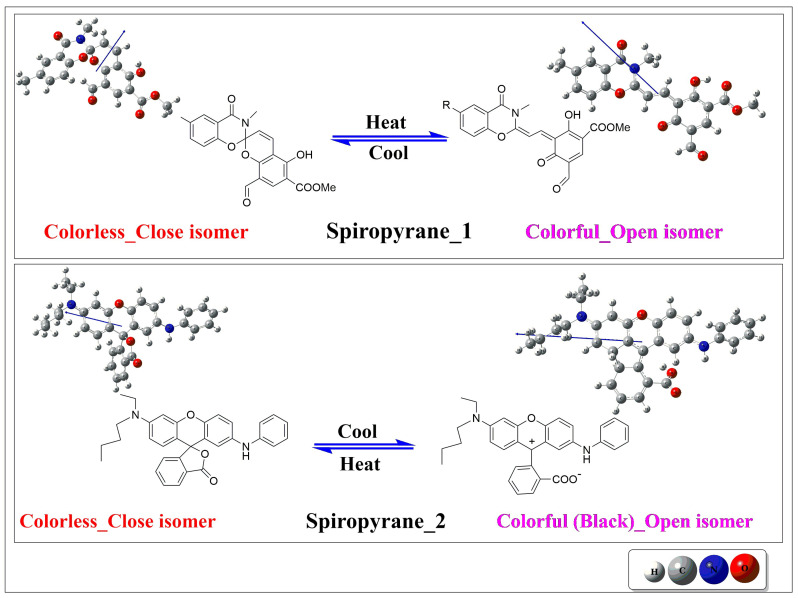
The close and open isomers of spiropyranes 1 [29] (top where R = Methyl group) & 2 (bottom) [30].

**Figure 2 molecules-28-06283-f002:**
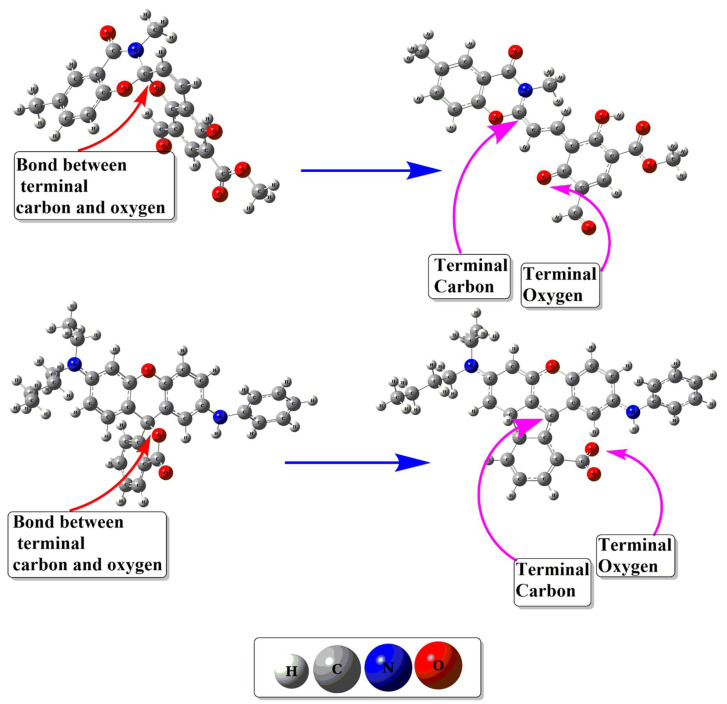
Optimized structures of close and open isomers of spiropyranes (**1** & **2**).

**Figure 3 molecules-28-06283-f003:**
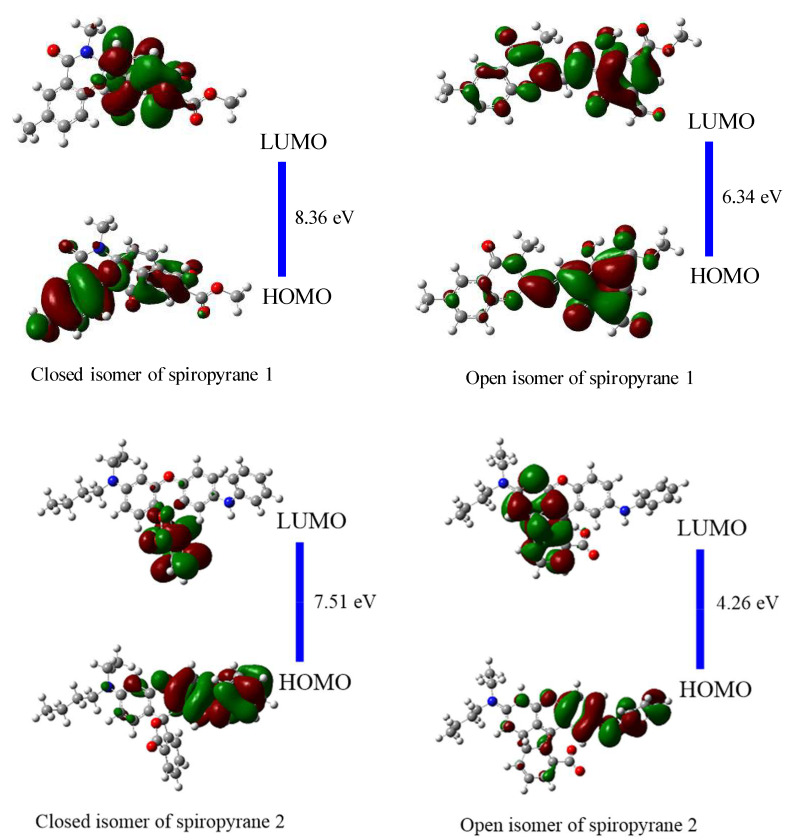
HOMOs-LUMOs densities and their gaps (in eV) of close and open isomers of spiropyranes (**1** & **2**).

**Figure 4 molecules-28-06283-f004:**
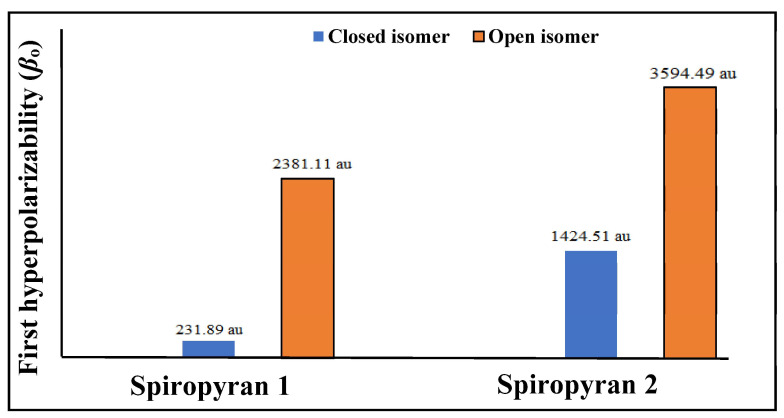
Graphical representation of hyperpolarizability *β*_o_ (in au) for close and open isomers of piropyranes (**1** & **2**).

**Figure 5 molecules-28-06283-f005:**
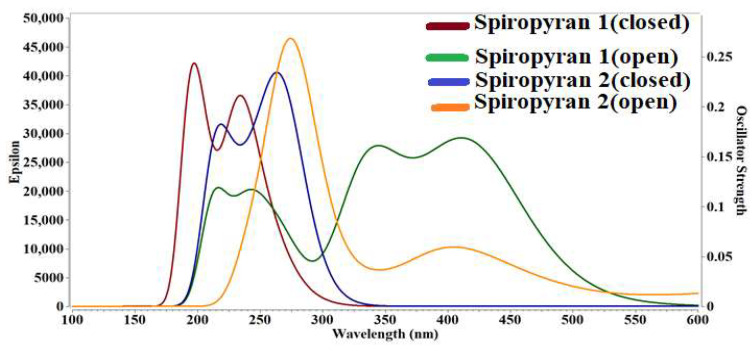
UV-Visible absorption spectra of closed and open isomers of spiropyranes (**1** & **2**).

**Figure 6 molecules-28-06283-f006:**
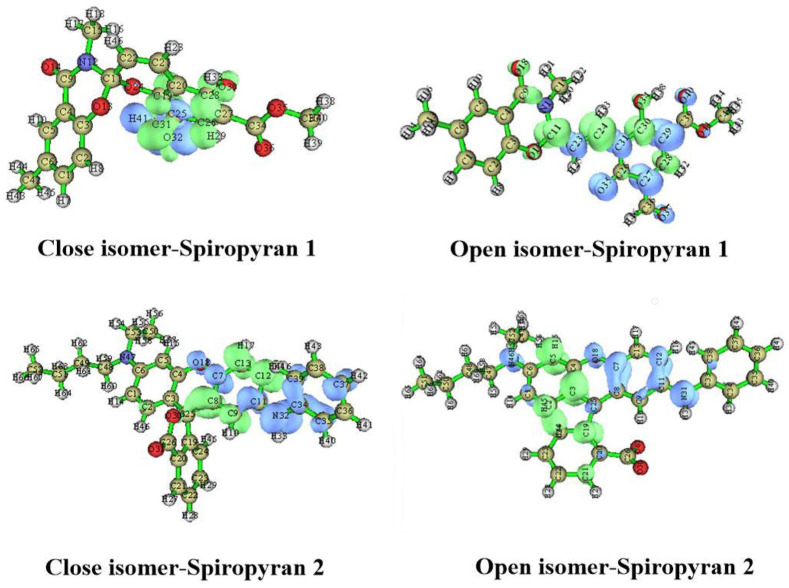
Density difference surfaces of closed and open isomers of spiropyranes (**1** & **2**) at excited state.

**Table 1 molecules-28-06283-t001:** Bond distance of ring-closing carbon and oxygen atoms (b_C-O_ in Å), activation energy (E_act_), relative energy of close and open isomer (E_rel_ in kcal mol^−1^) and charge (|e|) on ring-closing carbon and oxygen atoms of Spiropyran (**1** & **2**).

Compounds	Isomer	b_C-O_	E_rel_	E_act_	|e|	
					Carbon Atom	Oxygen Atom
Spiropyran **1**	close	1.43	0.00	12.74	0.75	−0.59
	open	4.11	5.78		0.64	−0.63
Spiropyran **2**	close	1.36	0.29	14.83	0.37	−0.57
	open	4.30	0.00		−0.05	−0.65

**Table 2 molecules-28-06283-t002:** The HOMOs energies (E_H_ in eV) and LUMO energies (E_L_ in eV), HOMO-LUMO energy gap (E_H-L_ in eV), ground state dipole moments (*µ*, in Debye), polarizability (*α*_o_, in au) and the first hyper polarizability (*β*_o_, in au) for close and open isomers of spiropyranes **1** & **2**.

Compounds	Isomer	E_H_	E_L_	G_H-L_	*µ*	*α_o_*	*β_o_*
spiropyran **1**	close	−8.74	−0.38	8.36	6.489	276	231.89
	open	−7.75	−1.41	6.34	8.783	350	2381.11
spiropyran **2**	close	−7.25	0.25	7.51	6.258	413	1424.51
	open	−9.61	−5.35	4.26	10.223	460	3594.49

**Table 3 molecules-28-06283-t003:** Transition energy (ΔE in eV) oscillator strength (ƒ_o_) maximum absorbance (λ_max_ in nm) and excited dipole moment (Δ*µ*) for close and open isomers of spiropyranes **1** & **2**.

Compounds	Isomer	ΔE	ƒ_o_	λ_max_	Δ*µ*
spiropyran **1**	close	6.40	0.54	194	6.488
	open	2.97	0.68	418	8.789
spiropyran **2**	close	4.62	0.41	269	6.259
	open	4.55	0.53	272	10.227

**Table 4 molecules-28-06283-t004:** Static first hyperpolarizability, frequency-dependent first hyperpolarizability coefficients in terms of electro-optic Pockel’s effect *β*_o_ (−ω; ω, 0) and second harmonic generation *β*_o_ (−2ω; ω, ω) (in au) for close and open isomers of spiropyranes **1** & **2**.

Compound	Isomer	Frequency	*β_o_* (0; 0, 0)	*β_o_* (−ω; ω, 0)	*β_o_* (−2ω; ω, ω)
Spiropyran **1**	close	0	231.9		
		5.63 × 10^14^ Hz (0.09 au)		359.6	4984.1
		2.82 × 10^14^ Hz (0.04 au)		255.9	317.9
	open	0	2373.2		
		5.63 × 10^14^ Hz (0.09 au)		6324.6	6698.7
		2.82 × 10^14^ Hz (0.04 au)		2971.4	5125.1
Spiropyran **2**	close	0	1424.5		
		5.63 × 10^14^ Hz (0.09 au)		2097.8	51,645.1
		2.82 × 10^14^ Hz (0.04 au)		1565.4	1893.9
	open	0	3594.5		
		5.63 × 10^14^ Hz (0.09 au)		22,805.2	54,124.8
		2.82 × 10^14^ Hz (0.04 au)		8774.1	23,919.6

**Table 5 molecules-28-06283-t005:** Static second-order hyperpolarizability (*γ*_o_ in au), frequency-dependent second-order hyperpolarizability coefficients in terms of dc-Kerr *γ*(−ω; ω,0,0) effect in au, EFSHG (in au) and refractive index (*n*_2_, in cm^2^/W) for close and open isomers of spiropyranes **1** & **2**.

Compounds	Isomers	Frequency (in Hz)	*γ*(0; 0, 0, 0)	*γ*(−ω; ω, ω, 0)	*γ*(−2ω; ω, ω, 0)	*n* _2_
Spiropyran **1**	close	0	3564.7			
		5.63 × 10^14^ Hz (0.09 au)		5095.1	3037.8	3.23 × 10^−18^
		2.82 × 10^14^ Hz (0.04 au)		3909.5	4636.6	1.01 × 10^−18^
	open	0	6907.9			
		5.63 × 10^14^ Hz (0.09 au)		52,842.8	2398.2	1.99 × 10^−17^
		2.82 × 10^14^ Hz (0.04 au)		9030.3	23,299.4	3.25 × 10^−18^
Spiropyran **2**	close	0	11,533.7			
		5.63 × 10^14^ Hz (0.09 au)		20,011.3	1027.6	1.11 × 10^−17^
		2.82 × 10^14^ Hz (0.04 au)		13,033.1	1713.9	3.45 × 10^−18^
	open	0	27,098.8			
		5.63 × 10^14^ Hz (0.09 au)		126,583.9	11,287.0	2.21 × 10^−17^
		2.82 × 10^14^ Hz (0.04 au)		120,760.7	33,722.1	1.50 × 10^−17^

**Table 6 molecules-28-06283-t006:** Transition energy (ΔE in eV) oscillator strength (ƒ_o_) maximum absorbance (λ_max_ in nm), excited dipole moment (Δ*µ* in Debye) ground state dipole moments (*µ*, in Debye), polarizability (*α*_o_, in au) and the first hyper polarizability (*β*_o_, in au) for close and open isomers of spiropyranes **1** & **2** in the presence of acetonitrile solvent.

Compounds	Isomer	ΔE	ƒ_o_	λ_max_	Δ*µ*	*µ*	*α_o_*	*β* _o_
spiropyran **1**	close	5.26	0.60	236	8.94	8.94	373	1242.20
	open	5.03	0.51	246	11.74	11.73	476	4876.61
spiropyran **2**	close	4.57	0.62	271	8.36	8.36	553	2844.07
	open	4.60	0.89	270	13.91	13.91	640	3578.72

**Table 7 molecules-28-06283-t007:** The integral of hole-electron (S) overlap and the distance between the centroid of hole and electron (D in Å) for close and open isomers of spiropyranes **1** & **2** to five excited states.

Compounds	Parameters	Ex. 1	Ex. 2	Ex. 3	Ex. 4	Ex. 5
spiropyran **1**-close isomer	S	0.21	0.51	0.45	0.53	0.17
	D	1.57	0.26	0.67	0.23	2.07
spiropyran **1**-open-isomer	S	0.34	0.15	0.39	0.20	0.27
	D	1.89	2.36	1.11	1.92	2.84
spiropyran **2**-close isomer	S	0.26	0.30	0.37	0.20	0.25
	D	1.94	1.77	0.97	2.51	1.64
spiropyran **2**-open isomer	S	0.35	0.21	0.25	0.34	0.37
	D	2.14	1.73	2.43	2.98	1.86

## Data Availability

Data will be made available to the corresponding author upon request.

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
