# Peer review of "Role of Delocalization, Asymmetric Distribution of π-Electrons and Elongated Conjugation System for Enhancement of NLO Response of Open Form of Spiropyran-Based Thermochromes"

_molecules, 2023, doi:10.3390/molecules28176283_

Round 1
Reviewer 1 Report
In this study, the authors used the DFT method to investigate the geometric, electronic, and nonlinear optical properties of spiropyranes thermochromes. The stability of selected spiropyranes closed and open rings was investigated using relative energies. Electronic properties were investigated through FMO analysis. A lower HOMO-LUMO energy gap and lower excitation energies were observed for the open ring spiropyrans, leading to large first hyperpolarizability values. π-electron resonance, asymmetric distribution, and elongated conjugated systems are the main factors for the higher hyperpolarizability of the open ring. In addition, the frequency-dependent hyperpolarizability and refractive index of the thermochromes considered were also analyzed; the NLO response increased significantly with increasing frequency. Among these compounds, the highest refractive index values were observed for the open ring of spiropyran 1. Molecular absorption analysis confirmed electronic excitation in the open ring compared to the closed ring. The results suggest that reversible thermochromic compounds can function as excellent NLO molecular switches and can be used to design advanced electronic devices. This paper will bring very useful information to researchers involved in organic devices. Therefore, I recommend that this manuscript be accepted for publication in Molecules.
Some minor comments are presented below.
1) One ketone group is hidden in Figure 1, and I recommend that the figure be corrected.
2) Does the structure of spiropyrane 2 in open form shown in Figure 1 correspond to that shown in Figure 2? In Figure 2, one can see a five-membered ring at the tip of the red arrow on the left side, but there is no such thing in the structural formula in Figure 1. The tip of the red arrow on the right side of Figure 2 is COOH, but in the structural formula of Figure 1, it is COO-.
3) In the open form of spiropyrane 2 in Fig. 1, what are the valence state or hybridization state of the carbon atom at the contact point of xanthene and benzoic acid? Is there hydrogen bonded there?
4) The energy change associated with the conformational change of spiropyrane is shown in Table 1, but what is the activation barrier?
5) The authors have missed a relevant reference.
https://doi.org/10.1021/jz401228c
Reviewer 2 Report
The paper Kosar et. al. Investigates the non linear optical (NLO) properties of two spiropyran derivatives. For each molecule two isomers are considered an open and a closed form. Various properties connected to NLO response, such as polarizability, first and second order static and frequency dependent hyperpolarizability, and absorption spectra, are calculated and analyzed, arriving at the conclusion that the closed-to-open conversion of these molecules could be potentially employed in NLO molecular switches.
While the computational methods employed are in principle a reasonable choice the results presented are not clear or in some cases even seem to be not correct. For example, in Figure 1 the isomerization of molecule 2 is seen to form a zwitterionic molecule. However, in Figure 2 the carboxylic moiety is protonated but there is no discussion of the reason of the choice of this particular structure compared to the zwitterion. The coordinates of the optimized structures are not provided and is therefore much more difficult to evaluate the structures that were chosen to represent the two isomers. Regarding the properties’ calculations, in Table 2 and Table 3 are reported the dipole moments of the optimized structures in the ground state and first excited state, respectively. However, the values are identical in the two tables (the same values are reported in the text) and I find it very unlikely that there is no change at all in the dipole moment upon excitation (there is no discussion on this apparent oddity in the text). There is a bit of confusion in the meaning of frequency and wavelength as in tables 4 and 5 the column “Frequency” reports correctly 0 for the static case but it then shows the wavelength for the other two cases. Finally, on page 10 the authors state that “Both isomers (close and open) of spiropyran 1 & 2 show transparency in deep UV region (below 200 nm)” while only a couple of lines before they wrote “It is found that the close isomer of the spiropyran 1 showed absorbance in UV region at 194 nm” which is again an indication of the poor quality of the presentation of the results.
Moreover, the main content of the paper reads as list of the computed data with mostly qualistative analyses, lacking a quantitative investigation of cause of the studied effects (e.g., calculation of the charge-transfer distance or density difference between ground and excited states) and an exhaustive explanation. Moreover, the cited literature to support the findings is only loosely related.
I therefore cannot recommend the publication of this work in Molecules.
Reviewer 3 Report
In this study, using density functional theory, authors have explored the geometric, electronic, optical, and nonlinear optical properties of reversible thermochromic spiropyranes.
I see the potential and motivation behind these analyses, and it will be worth being published in Molecules after some revisions pointed out as follows.
1. It seems the authors have conducted gas phase calculations(s). Generally, optical characteristics and photophysical parameters are affected by solvent polarities. Can authors explain the logic behind performing only ‘gas-phase’ calculations?
2. To elucidate the origin of electronic spectra, TD-DFT computations were performed at the same level of theory. I would encourage authors if they can provide some results by using the polarizable continuous solvation technique PCM, PCM-TD-DFT.
3. In section 3.1 authors mentioned that ‘optimized geometries are given in Fig 1’, however, it is a sketch or more like a ChemDraw-generated figure. Authors should give an actual figure of their optimized geometries and the vector of dipole moment shown in the figure.
4. NBO calculation is an essential part of this study, but the authors did not include the original citation of the NBO paper (1983 paper).
Round 2
Reviewer 2 Report
Thank you for providing the updated version of the manuscript and answering all my concerns. The new version is definitely improved and the analysis of the excited states gives a better view of the systems but in my opinion there are still two points that should be still addressed.
1) “Indeed, these results are surprising. The values of dipole moments have very minute differences (in the third decimal place) in the gas phase however the values shown are with two decimal places. The dipole moments values change significantly between ground and excited state when calculations are performed in acetonitrile. The dipole moment values in solvent have been added in the supplementary information.”
I still find it puzzling that the values are exactly the same after an electronic excitation, i.e. rearrangement of the electron density but if the authors are completely sure to have used the excited state density (and not the ground state one as it is the default in Gaussian) I would advise to add a short sentence in the text to confirm the soundness of the results.
2) “In previous literature, frequency is reported in dynamic hyperpolarizability, but the values are given in the units of wavelength (nm). Somehow this convention is adopted in the literature. Secondly, the zero nm frequency is mentioned for static hyperpolarizability while for dynamic hyperpolarizability we used two most commonly use frequencies (532 and 1064 nm) in laser technology.”
I still find it confusing having the mixed units in the same table, since the wavelengths can easily be converted into frequency values.
The analysis added in the revised version of the manuscript could benefit by some polishing of the English language.
Reviewer 3 Report
The authors have addressed all the issues I had written in my comments and the quality of the manuscript have improved. So, I recommend this to be accepted in 'Molecules'.
Author Response
Response to the Comment of Reviewer 3
Manuscript ID: molecules-2457680
Title: Role of delocalization, asymmetric distribution of π-electrons and elongated conjugation system for enhancement of NLO response of open form of Spiropyran based thermochromes
Comment:
The authors have addressed all the issues I had written in my comments and the quality of the manuscript have improved. So, I recommend this to be accepted in 'Molecules'.
Response: We are very grateful to our reviewer for accepting our manuscript to be published in its current form.